# Pancytopenia in a Case of Aplastic Anaemia/Paroxysmal Nocturnal Haemoglobinuria Unmasked by SARS-CoV-2 Infection: A Case Report

**DOI:** 10.3390/medicina58091282

**Published:** 2022-09-15

**Authors:** Arcangelo Iannuzzi, Antonio Parrella, Francesca De Ritis, Anna Cammarota, Lucia Berloco, Francesca Paudice, Giovanni D’Angelo, Emilio Aliberti, Gabriella Iannuzzo

**Affiliations:** 1Department of Medicine and Medical Specialties, A. Cardarelli Hospital, 80131 Naples, Italy; 2Department of Acute Internal Medicine, Royal Victoria Infirmary, Newcastle NE1 4LP, UK; 3Department of Clinical Medicine and Surgery, Federico II University, 80131 Naples, Italy

**Keywords:** COVID-19, Paroxysmal Nocturnal Haemoglobinuria, PIGA gene, aplastic anaemia

## Abstract

During an acute SARS-CoV-2 infection, a diagnosis of Aplastic Anaemia associated with Paroxysmal Nocturnal Haemoglobinuria (AA/PNH) was made in a 78-year-old woman who had presented to the emergency department with severe pancytopenia. It is possible that she had subclinical AA/PNH that was unmasked during the acute COVID-19 infection, but we can also suspect a direct role of the virus in the pathogenesis of the disease, or we can hypothesize that COVID-19 infection changed the phosphatidylinositol glycan class A (PIGA) gene pathway.

## 1. Introduction

SARS-CoV-2, an airborne virus first identified in 2019, spread rapidly on a global level, causing the COVID-19 pandemic. It has been shown that this virus is able to activate complement, both directly and indirectly, creating a state of generalised inflammation [1,2]. In this article we present a possible relationship between SARS-CoV-2 infection and the appearance of signs and symptoms of aplastic anaemia in a patient who was a carrier of the Paroxysmal Nocturnal Haemoglobinuria (PNH) clone, a condition of which she was not previously aware.

## 2. Case Report

We present the case of a 78-year-old Caucasian female, who presented to the Emergency Department after being found to be severely anaemic (6.4 g/dL Hb) whilst being investigated as an outpatient for asthenia and fatigue.

Her past medical history included: stage four chronic kidney disease, arterial hypertension, hypercholesterolaemia, and previous genital carcinoma. She had not been vaccinated against SARS-CoV-2.

On examination she was alert and responsive, eupnoeic at rest, with pale skin and mucosal membranes, and no gross motor deficits. She had a blood pressure of 140/80 mmHg, a heart rate of 74 bpm, SpO2 at 99% in air, and was apyrexial. Her Glasgow Coma Scale score was 15. She presented a normal objective neurological examination. A thoracic examination revealed normal vesicular breath sounds occurring bilaterally in all regions with no abnormal sounds. An abdominal examination showed that the abdomen was soft, not tender on both superficial and deep palpation, with no masses palpable. She had no peripheral oedema.

A rapid SARS-CoV-2 swab taken on admission was found to be positive, and this was confirmed with molecular testing. Following this, the patient was isolated until the infection had resolved.

Admission blood tests showed 6.0 g/dL Hemoglobin (Hb), 2.17 × 10^6^ Red Blood Cells, 1.5/1000 Reticulocytes, 1.19 × 10^3^ White Blood Cells (10.6% Granulocytes, 86.8% Lymphocytes, 2.1% Monocytes, 0.5% Eosinophils, 0% Basophils), 3 × 10^3^ Platelets, 3.05 mg/dL creatinine, 0.45 mg/dL total bilirubin, 193 IU/dL lactate dehydrogenase (LDH), Anti-Nuclear Antibodies (ANA) at a ratio of 1:320 with a homogenous pattern, and was both anti-dsDNA- and Extractable Nuclear Antigen (ENA)-negative. A peripheral blood smear showed red cell anisopoikilocytosis.

Computed tomography (CT) of the head showed chronic hypoperfusion. CT with contrast of the thorax and abdomen showed a pericardial effusion, a slightly enlarged liver with no focal lesions, a slightly thickened gallbladder with no stones, and kidneys in situ but slightly reduced in size, with all remaining findings normal.

During admission, this patient required multiple transfusions of red cells and platelets.

Once the patient tested negative by PCR/Antigen testing, it was possible to perform needle aspiration of the bone marrow, which revealed evidence of sufficient cellularity with abundant stroma, granular and erythroid series represented at all stages of maturation with dysplastic aspects, rare megakaryocytes, an absence of blasts and cells of extra-hemopoietic origin, and numerous small lymphocytes.

Based on investigations already performed and the presence of aplastic anaemia without blasts or cells of extra-haematopoietic origin, PNH was suspected, even in the absence of thrombotic events or episodes of haematuria. Cytofluorometric analysis was requested, which was positive for the PNH clone (Table 1).

To confirm suspected medullary aplasia, an osteomedullary biopsy was performed. This sample showed poor cellularity. Cytometric analysis demonstrated the presence of cells defective in glycophosphatidylinositol (GPI)-linked molecules in the population of granulocytes and monocytes. Immuno-cytometric analysis with monoclonal antibodies showed: poor cellularity, a maturation curve of the granulocyte population with line alterations (with a maturation block at the level of the intermediate myeloid compartment), a monocytic population equal to 2% of the cells, a lymphoid component equal to 37% of the cells, and an absence of CD34+ cells.

During the diagnostic–therapeutic process, the patient’s clinical condition progressively deteriorated, with the patient being voluntarily discharged before it was possible to initiate any specific treatment for this disease. The deteriorating clinical condition and the sudden decision to leave the hospital during the night did not allow us the opportunity to request consent for the publication of this clinical case; moreover, the patient and her family did not respond when attempts were made to contact them with the details previously provided.

## 3. Discussion

PNH is an acquired haematological disorder caused by mutation of the PIGA gene present on the X chromosome [3].

The estimated incidence of PNH is around 15.9 cases per million people globally, although this number could be underestimated due to unrecognised or asymptomatic cases. Incidence appears to be equal between sexes and the median age of presentation is around 30 years of age [4]. AA/PNH has a bimodal age distribution, with the median age of presentation being around 60 years of age.

The PIGA gene codes for an enzyme implicated in the synthesis of the GPI anchor, a molecule which anchors different proteins to the cell membrane of haematopoietic stem cells. Mutated cells multiply, creating clones which will have partially functional or totally inactive enzymes, depending on the mutation. There are many proteins which use GPI as an anchor on the cell membrane. Of these, those that possess a key role in the pathogenesis of PNH are the inhibitors of complement CD55, which stabilises C3-C5 convertase, and CD59, which inhibits the formation of the membrane attack complex (MAC) [5]. The loss of these molecules from the erythrocytic membrane makes these cells more susceptible to complement attack (through constant low-level activation of the alternative pathway), and cell lysis (triggered by infections or other events). In this way, the patient will encounter a basal level of haemolysis and paroxysmal haemolytic attacks [3]. 

Clinically, this disease can present in a variable manner, creating difficulty in arriving at the diagnosis. The International PNH Interest Group (IPIG) has proposed a clinical classification of the disease which distinguishes between: classical PNH, characterised by haemolysis and thrombosis; PNH associated with medullary pathology, such as aplastic anaemia and myelodysplastic syndromes, usually with small clones; and a subclinical form, with small clones and no evidence of haemolysis in laboratory investigation [6].

Intravascular haemolysis leads to changes in smooth muscle tone, with muscular spasm, vasoconstriction, and platelet activation, which can drive thrombotic phenomena [7]. Thromboses in PNH are more commonly seen in the venous system and often occur in atypical locations, for example, in hepatic (Budd–Chiari syndrome), cerebral, mesenteric, portal, splenic and renal veins [5]. Recently we published the case of a patient with PNH who presented with haemolytic anaemia and venous thrombosis in atypical locations [8]. Other complications of this disease include the development of both acute and chronic renal impairment due to tubular damage caused by iron deposition, free haemoglobin in plasma and microvascular thrombosis [9].

PNH associated with medullary disorders such as aplastic anaemia and myelodysplastic syndrome (as is the case presented by us) is characterised by leukopenia, thrombocytopenia, and a low reticulocyte count, with a predisposition to haemorrhage and a tendency to develop infections. Obtaining an early diagnosis is essential to promptly intervene and avoid clinical complications associated with the disease [10].

Interestingly, a correlation between SARS-CoV-2 infection and PNH appears to be emerging. In fact, from the beginning of the pandemic, many cases of clinical deterioration and increased haemolysis in patients with a known diagnosis of PNH following infection with SARS-CoV-2 have been reported [11,12,13,14,15,16,17,18,19,20,21]. There is one case report of a COVID-19 infection which presented with deep vein thrombosis in a patient with PNH [20]. Hines et al. have described a case in which a patient received a diagnosis of PNH during SARS-CoV-2 infection, without typical symptoms and in which, considering the temporal correlation, it appears possible that there was a direct role of the virus in the pathogenesis of the disease [21]. Table 2 includes all cases reported in the scientific literature on the relationship between COVID-19 infection and PNH. Other studies have also documented haemolysis induced by SARS-CoV-2 mRNA vaccination in patients with PNH [22,23].

This virus is able to activate complement directly and indirectly. The virus’ envelope proteins bind to the MBL (mannose-binding lectin) protein, inducing the lectin pathway. Furthermore, antibodies specific for SARS-CoV-2 form complexes with C1q, activating the classical complement pathway. Eventually, spike proteins compete with factor H (a C3 inhibitor) in binding with heparan sulphate; in this way, the inhibitory effect of factor H on C3 is decreased, favouring the activation of the alternative complement pathway [2].

Treatment of PNH should be considered in two principal clinical presentations: (1) haemolytic PNH without overt marrow failure and (2) moderate/severe aplastic anaemia or PNH with marrow failure. Treatment options for haemolytic PNH have been revolutionised by the introduction of eculizumab, a monoclonal antibody targeting component 5 (C5) of the complement cascade. Subsequently, many other novel anti-complement agents have been developed and are now in clinical development for haemolytic PNH. On the other hand, PNH associated with aplastic anaemia and pancytopenia, as occurred in the case described in the present manuscript, requires a different treatment regime. For these patients, therapy should target the bone marrow failure with either bone marrow transplantation (if <40 years old) or immunosuppression [24]. The outcome of patients >40 years old with AA/PNH is poor, as seen in a large UK cohort where only 16.6% showed disease recovery (as defined by improved blood count, with or without reduction of the EPN clone) [25].

## 4. Conclusions

To our knowledge, the present case report represents the first case of isolated pancytopenia in the setting of aplastic anaemia/PNH presenting during an acute COVID-19 infection in Italy. We cannot determine whether COVID-19 infection expanded an already existing subclinical PIGA gene mutation, just as we cannot exclude the possibility that the occurrence of both SARS-CoV-2 infection and AA/PNH manifestation was coincidental. PNH is a rare disease, but the finding of pancytopenia in the course of a viral infection, especially a SARS-CoV-2 infection, should increase clinical suspicion of an aplastic anaemia associated with the PNH clone and therefore urge prompt performance of a flow cytometric examination. In addition, these patients require clinical and laboratory surveillance because, although it is rare, they sometimes develop haemolytic or thrombotic manifestations due to PNH clone expansion.

## Figures and Tables

**Table 1 medicina-58-01282-t001:** PNH flow cytometry.

Population	N° Events in the Gate	Sensibility % (LOD)	Gating Markers	GPI-Linked Markers	% Negative (PNH III)	% Defective (PNH II)	% Normal (PNH I)	Clone Size (PNH II + PNH III)
Granulocytes	4520	0.66%	CD45, CD15	FLAER	93.5%	2.5%	4%	6.5%
Monocytes	600	5.00%	CD45, CD33	FLAER	93.5%	2.5%	4%	6.5%

Legend: LOD: limit of detection; PNH III: complete deficiency of GPI-linked proteins, PNH type III cells; PNH II: partial deficiency of GPI-linked proteins, PNH type II cells; PNH I: normal presence of GPI-linked proteins, PNH type I cells; FLAER: fluorescent aerolysin. Modified from Rotoli B.

**Table 2 medicina-58-01282-t002:** Clinical presentation of COVID-19 in patients with PNH.

Authors	PNH Type	Clinical Presentation	History of PNH	Therapy
Kulasekararaj et al. [11]	2 Classical PNH; 2 AA/PNH	Pt 1: F, 35 yrs, fever, sore throat, myalgia	Long-standing history (6 yrs) of classical PNH (eculizumab, then ravulizumab)	Pt 1: Ravulizumab
		Pt 2: M, 47 yrs, fever, abdominal pain, nausea, cough	Naive	Pt 2: Oxygen, Warfarin
		Pt 3: F, 37 yrs, fever, headache, cough, myalgia	Long-standing history (11 yrs) of AA/PNH (eculizumab)	Pt 3: Eculizumab, Blood transfusion
		Pt 4: M, 51 yrs, fever, cough, abdominal pain, fatigue, myalgia	No specified history of AA/PNH	Pt 4: Oxygen, Warfarin
Fattizzo et al. [12]	1 Classical PNH	Shortness of breath, asthenia, dark urine, anaemia	Long-standing history (11 yrs) of classical PNH (eculizumab)	Antibiotics, Dexamethasone, LMWH, Eculizumab, Red blood cell transfusion
Otieno et al. [13]	AA/PNH	F, 19 yrs, tooth infection, pancytopenia and haemolysis	Naive	Eculizumab and then Ravulizumab
Sokol et al. [14]	1 Classical PNH	M, 27 yrs, acute onset shortness of breath, cough, blood in the urine	Naive	LMWH, Dexamethasone, Cefuroxime
		Pt 2: M, 47 yrs, fever, myalgia, dizziness	Pt 2: Long-standing history (6 yrs) of AA/PNH (eculizumab), recently MDS-transformed and awaiting BMT	Pt 2: Antibiotics.
		Pt 3: M, 43 yrs, breakthrough haemolysis with haemoglobinuria, respiratory failure	Long-standing history (17 yrs) of AA and haemolytic PNH (14 yrs), (eculizumab)	Pt 3: Eculizumab, Packed red cells, Antibiotics, Mechanical ventilation. Death after 23 days.
		Pt 4: F, 77 yrs, breathlessness and chest pain	Long-standing history (15 yrs) of AA, and symptomatic PNH (9 yrs) (eculizumab)	Pt 4: Antibiotics, Packed red cells, Eculizumab.
Pike et al. [15]	1 Classic PNH 3 AA/PNH	Pt 1: F, 61 yrs, fever, lethargy, dry cough, diarrhoea	Pt 1: Long-standing history (12 yrs) of classical PNH (ravulizumab)	Pt 1: Oxygen and Antibiotics.
		Pt 2: M, 47 yrs, fever, myalgia, dizziness	Pt 2: Long-standing history (6 yrs) of AA/PNH (eculizumab), recently MDS transformed and awaiting BMT	Pt 2: Antibiotics.
		Pt 3: M, 43 yrs, breakthrough haemolysis with haemoglobinuria, respiratory failure	Long-standing history (17 yrs) of AA and haemolytic PNH (14 yrs), (eculizumab)	Pt 3: Eculizumab, Packed red cells, Antibiotics, Mechanical ventilation. Death after 23 days.
		Pt 4: F, 77 yrs, breathlessness and chest pain	Long-standing history (15 yrs) of AA, and symptomatic PNH (9 yrs) (eculizumab)	Pt 4: Antibiotics, Packed red cells, Eculizumab.
Barcellini et al. [16]	4 Classical PNH	Pt 1: F, 55 yrs, fever, ageusia, active haemolysis	Pt 1: Naive	Pt 1: Prophylactic LMWH, Azithromycin
		Pt 2: F, 58 yrs, no new symptoms	Pt 2: Long-standing history (14 yrs) classical PNH (eculizumab, then ravulizumab)	Pt 2: Ravulizumab, Warfarin
		Pt 3: F, 41 yrs, no new symptoms	Pt 3: Unspecified history of therapy with ravilizumab	Pt 3: Ravulizumab,
		Pt 4: M, 39 yrs, fever	Pt 4: Unspecified history of therapy with eculizumab	Pt 4: Eculizumab
Araten et al. [17]	2 Classical PNH	Pt 1: F, 39 yrs, fever, fatigue	Pt 1: Long-standing history (19 yrs) classical PNH (eculizumab, then ravulizumab; fondaparinux, cyclosporine)	Pt 1: Ciprofloxacin
		Pt 2: F, 60 yrs, cough, fatigue, malaise, headache	Pt 2: Long-standing history (15 yrs) classical PNH (eculizumab)	Pt 2: Eculizumab
Schuller et al. [18]	1 Classical PNH	F, 68 yrs, mild fever, cough, sore throat, diarrhoea, haemolytic crisis, mild pancytopenia	Long-standing history (45 yrs) classical PNH, (eculizumab started in 2007)	Eculizumab, Red cell concentrates, Granulocyte colony stimulating factor, anti-inflammatory agents
Genthon et al. [19]	1 Classical PNH	M, 45 yrs, dry cough, myalgia, diarrhoea, fever	Long-standing history (27 yrs) classical PNH (eculizumab started in 2008)	Amoxicillin/clavulanate, Ciprofloxacin, O_2_ therapy, Hydroxychloroquine, Ritonavir + Lopinavir, Enoxaparin, Eculizumab, Tocilizumab
Pravdic et al. [20]	1 Classical PNH	M, 33 yrs, DVT left popliteal vein, fever	Long-standing history (7 yrs) classical PNH (eculizumab and warfarin started in 2015)	Amoxicillin/clavulanate, Ciprofloxacin, LMWH, Eculizumab
Hines et al. [21]	1 AA/PNH	M, 35 yrs, pancytopenia, melena, headache, fatigue, poor appetite	Naive	Packed red blood cells transfusion, steroid, intravenous immunoglobulin, eculizumab

## Data Availability

Not applicable.

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
