# Peer review of "Pancytopenia in a Case of Aplastic Anaemia/Paroxysmal Nocturnal Haemoglobinuria Unmasked by SARS-CoV-2 Infection: A Case Report"

_medicina, 2022, doi:10.3390/medicina58091282_

Round 1

Reviewer 1 Report

Iannuzzi et al presented a case repot for a Caucasian female who had symptomatic PNH and SARS-CoV2 positive. They concluded that SARS-CoV2 infection could induce PNH manifestation as the women did not know she had PNH or PIGA gene mutation before SARs-CoV2 infection.  

Major points:

1. Although clinical tests showed PNH, the mutation  could have been confirmed by a simple PCR and sequencing of the PIGA gene or region of mutation.

2. The occurrence of both SARs-CoV2 infection and PNH manifestation also could be coincidental.

Minor comments:

1. In abstract, "but we also.......PIGA gene mutation" could be deleted. Because neither mutation is confirmed nor SARS-CoV2 does have ability to change DNA sequence or expand a mutation. But it can change the PIGA gene pathway to manifest the PNH symptoms.

2.  In page 2, from the sentence "Once the patient had become seronegative it was possible to proceed perform ........." , "proceed" should be omitted.

3. In page 3, before discussion, GPI should be abbreviated when it was first time described (GPI-linked...)

4. In page 3, in discussion, this sentence "mutation can be present in erythrocytes, leukocytes, platelets, lymphocytes." should be deleted. Because if there is a mutation it is present in all cells in the body but could express in specific cells or in tissues.

5. Figure 1 quality should be improved.  Legend should mention more detail and could be improved.

6. How was the patient consent waived? An explanation should be mentioned in the text.

Author Response

Comments and Suggestions for Authors: Reviewer 1

Iannuzzi et al presented a case repot for a Caucasian female who had symptomatic PNH and was SARS-CoV2 positive. They concluded that SARS-CoV2 infection could induce PNH manifestation as the women did not know she had PNH or PIGA gene mutation before SARs-CoV2 infection.  

Major points:

  1. Although clinical tests showed PNH, the mutation could have been confirmed by a simple PCR and sequencing of the PIGA gene or region of mutation.

The reviewer is right and it would have been correct to perform a PCR and sequencing of the PIGA gene; however, the consultant hematologist felt that flow cytofluorimetry and osteomedullary biopsy were sufficient to confirm the diagnosis and did not give the necessary consent to practice for genetic testingHowever, we modified the title of the case report in “Pancytopenia in a Case of Aplastic Anaemia / Paroxysmal Nocturnal Haemoglobinuria Unmasked by SARS CoV2 Infection: A Case Report”.

  1. The occurrence of both SARs-CoV2 infection and PNH manifestation also could be coincidental.

We have added in the conclusions the possibility that the occurrence of both SARS-CoV2 infection and PNH manifestation could have been coincidental.

Minor comments:

  1. In abstract, "but we also.......PIGA gene mutation" could be deleted. Because neither mutation is confirmed nor SARS-CoV2 does have ability to change DNA sequence or expand a mutation. But it can change the PIGA gene pathway to manifest the PNH symptoms.

We followed the advice of the reviewer.

  1. In page 2, from the sentence "Once the patient had become seronegative it was possible to proceed perform ........." , "proceed" should be omitted.

We apologize for the mistake. The word “proceed” has been deleted.

  1. In page 3, before discussion, GPI should be abbreviated when it was first time described (GPI-linked...)

We followed the reviewer's instructions.

  1. In page 3, in discussion, this sentence "mutation can be present in erythrocytes, leukocytes, platelets, lymphocytes." should be deleted. Because if there is a mutation it is present in all cells in the body but could express in specific cells or in tissues.

We agree with the reviewer and deleted the sentence.

  1. Figure 1 quality should be improved.  Legend should mention more detail and could be improved.

We followed the advice of the reviewer.

  1. How was the patient consent waived? An explanation should be mentioned in the text.

We specified in the text that the deteriorating clinical condition with the sudden decision to leave the hospital during the night did not allow us the opportunity to request consent for the publication of this clinical case; moreover, the patient and her family did not respond when attempts were made to contact them with details previously provided.

Reviewer 2 Report

In this report, Authors reported a case of possible pancytopenia in course of Paroxysmal Nocturnal Hemoglobinuria in a patient with SARS-CoV2 Infection.

Overall, the case is interesting, however the manuscript needs to be revised before publication.

Title: I think that the title should be rephrased since the diagnosis has been supposed by the Authors, but the definitive cause is undocumented.

Manuscript:

-        Please decodify PNH the first time in Introduction

-      The English should be revised by a native speaker, it is incorrect in same cases, while in others it is not fluent.

-        Page 2: Once the patient become seronegative. This is incorrect, the patient become negative to PCR/Antigen testing, but not seronegative (she probably became seropositive instead!)

-        Figure 1 should be modified; it is not completely readable. Moreover, since it is not in English, it requires Legend to be correctly read.

-   The discussion should include the main message that Authors want to tell other colleagues and not just the explanation of the case. What is the clinical lesson? What should be done in similar cases? Etc.

-        Ethical statement should be included.

Additional comments:

-        In what phase of pandemic this case occurred? Could authors verify if other cases occurred in the same period? This could be associated with Variants of Concern.

-        A Table including all cases reported in literature may be useful for readers.

Author Response

Comments and Suggestions for Authors: Reviewer 2

In this report, Authors reported a case of possible pancytopenia in course of Paroxysmal Nocturnal Hemoglobinuria in a patient with SARS-CoV2 Infection.

Overall, the case is interesting, however the manuscript needs to be revised before publication.

Title: I think that the title should be rephrased since the diagnosis has been supposed by the Authors, but the definitive cause is undocumented.

We have changed the title “Pancytopenia in a Case of Aplastic Anaemia / Paroxysmal Nocturnal Haemoglobinuria Unmasked by SARS CoV2 Infection: A Case Report.”

Manuscript:

-      Please decodify PNH the first time in Introduction

We followed the advice of the reviewer.

-      The English should be revised by a native speaker, it is incorrect in same cases, while in others it is not fluent.

       The manuscript has been revised by Dr. Emilio Aliberti (a co-author of the article), who is a native English speaker.

-      Page 2: Once the patient become seronegative. This is incorrect, the patient become negative to PCR/Antigen testing, but not seronegative (she probably became seropositive instead!)

       The reviewer is right. We reworded the paragraph.

-      Figure 1 should be modified; it is not completely readable. Moreover, since it is not in English, it requires Legend to be correctly read.

We followed the advice of the reviewer.

-      The discussion should include the main message that Authors want to tell other colleagues and not just the explanation of the case. What is the clinical lesson? What should be done in similar cases? Etc.

We added in the Discussion and Conclusions the following sentences:

DISCUSSION…… Treatment of PNH should be considered in two principal clinical presentations: 1) haemolytic PNH without overt marrow failure and 2) moderate/severe aplastic anaemia or PNH with marrow failure. Treatment options for haemolytic PNH have been revolutionised by the introduction of eculizumab, a monoclonal antibody targeting component 5 (C5) of the complement cascade. Subsequently, many other novel anti-complement agents have been developed and are now in clinical development for haemolytic PNH. On the other hand, PNH associated with aplastic anaemia and pancytopenia, as the case described in the present manuscript, requires a different treatment regime. For these patients, therapy should target the bone marrow failure with either bone marrow transplantation (if <40 years old) or immunosuppression.24 The outcome of patients >40 years old with AA/PNH is poor, seen in a large UK cohort where only 16.6% showed disease recovery (as defined by improved blood count, with or without reduction of the EPN clone).

CONCLUSIONS: ……. we cannot exclude that the occurrence of both SARS-CoV2 infection and AA/PNH manifestation could have been coincidental. PNH is a rare disease but the finding of pancytopenia in the course of a viral infection, especially a SARS-CoV2 infection, should increase clinical suspicion of an aplastic anaemia associated with PNH clone and therefore urge prompt performance of a flow cytometric examination. In addition, these patients require clinical and laboratory surveillance because, although rarely, they sometimes develop haemolytic or thrombotic manifestations due to PNH clone expansion.

-      Ethical statement should be included.

We are awaiting for the judgment of the ethics committee

Additional comments:

-      In what phase of pandemic this case occurred? Could authors verify if other cases occurred in the same period? This could be associated with Variants of Concern.

This case occurred in March 2022. We are not aware of similar cases in the scientific literature in the same period.

-      A Table including all cases reported in literature may be useful for readers.

We followed the advice of the reviewer and added Table 1

Round 2

Reviewer 2 Report

All comments have been correctly addressed.

Congratulations!